# Growing Networks by Folding Manifolds at Mistakes

## Abstract

Modern deep learning paradigms heavily rely on over-parameterized models, leading to excessive costs and limited interpretability. While growing neural networks (GrowNNs) offer a biologically inspired alternative by incrementally expanding architectures, existing methods lack theoretical grounding and often result in unstable, heuristic-driven growth. This paper proposes a novel geometric framework that interprets neural network growth as folding the learned representation manifolds to enhance model capacity. We theoretically establish that strategically adding neurons—equivalent to introducing geometric folds—at locations corresponding to systematic prediction mistakes optimally increases expressivity. Our method introduces: (1) A manifold-based strategy for effective network growth by identifying "typical mistakes" via clustering of mis-predictions and targeted folding; (2) A stable fine-tuning solution using gradient-aligned initialization and folding hyperplane regularization to ensure targeted correction of mistakes; (3) Ante-hoc instance-level interpretability, where each grown neuron can be justified and explained by a specific mis-predicted data instance representing a model deficiency. Experiments on synthetic manifolds, MNIST, and CIFAR-10 demonstrate controlled capacity expansion, competitive parameter efficiency, and inherent explainability throughout the growth process.

## 1 Introduction

Motivated by neural scaling laws (Kaplan et al., 2020; Isik et al., 2024), the modern deep learning paradigm tends to initialize over-parameterized models to ensure sufficient representational capacity before training. This practice has driven the excessive scaling up of models in both academia and industry, particularly with the dominance of Large Language Models (LLMs) (OpenAI, 2022; Guo et al., 2025), which typically contain far more parameters than necessary to fit their training data or to learn task-specific knowledge (Zhang et al., 2017; Thompson et al., 2020; Nakkiran et al., 2021; Aghajanyan et al., 2021).

Numerous studies have explored parameter-efficient approaches in deep learning. For instance, Neural Architecture Search (NAS) (Ren et al., 2021), Network Pruning (Cheng et al., 2024), and Dynamic Neural Networks (DyNNs) (Han et al., 2021) are typical research directions that adaptively adjust model architectures to enhance parameter efficiency. However, these methods often involve computationally intensive searches over massive candidate architectures, require distinct stages before or after the main training phase, or perform structural modifications mainly at the macro-level of blocks, layers, or modules instead of fine-grained neuron- or connection-level adaptations.

A promising line of research proposes incrementally growing small neural networks into larger, capable ones during training (Wu et al., 2019; Evci et al., 2022). This idea of growing neural networks (GrowNNs) is intuitively appealing: biological nervous systems learn mainly by growing connections (synapses) through interactions with the environment. Compared to the dominant directions described above, GrowNNs offer a more direct, efficient, and structurally coherent path to achieving optimized neural architectures [1].

---

[1] A detailed review of studies on the optimization of neural network architecture, along with related works on GrowNNs, is provided in the Appendix A.1.

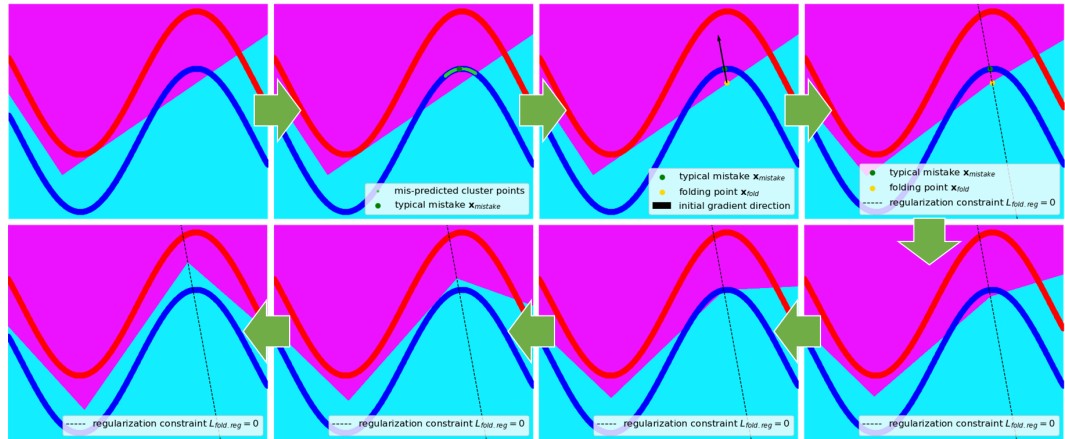

Figure 1: Visualization of the neural network's growing process on sine functions classification simulation dataset.

Nevertheless, all the approaches discussed above are primarily empirical, offering limited theoretical explanations for why specific architectural modifications optimize efficiency or how they influence learning dynamics. Particularly for GrowNNs, where current works rely heavily on intuition and focus mainly on the empirical efficiency of proposed growth policies (in a bid to rival NAS methods) without understanding how growth processes affect model capacity and learning behaviors, nor theoretically justifying why individual growth steps are effective and non-trivial. This often results in aimless growth and even unstable performance [2].

In this study, we provide an interpretation of the neural network growth process through the lens of established theories on manifold representations and expressivity of neural network architectures. Specifically, we posit that: **neural networks grow to enhance model capacity through geometrically "folding" the learned representation manifold**. This growth of capacity increases expressive power, enabling the potential correction of systematic prediction mistakes in the original, under-capacity model. Therefore, we propose that: **effective growth can be achieved by strategically adding "folds" at manifold regions corresponding to the model's "typical mistakes"**. We support this idea with theoretical analysis (Section 2), design a corresponding GrowNNs framework (Section 3), and validate its efficacy through both observing according phenomena in simulations on synthetic datasets as well as achieving successful learning on large scale images datasets like MNIST and CIFAR-10 (Section 4).

In summary, this paper makes the following contributions:
- We provide a novel geometric interpretation of the neural network growing process.
- We propose an advanced GrowNNs framework featuring:
  - An explainable, targeted growing strategy that identifies "typical mistakes" via mis-prediction clustering and conducts targeted "manifold folding";
  - A stable fine-tuning solution using gradient-aligned initialization and folding hyperplane regularization to ensure targeted correction of mistakes;
  - Ante-hoc instance-level interpretability, where each grown neuron can be justified and explained by a specific mis-predicted data instance representing a model deficiency.

## 2 THEOREMS & DEFINITIONS

### 2.1 GROWING NEURAL NETWORKS AS FOLDING MANIFOLD

Unlike XAI studies for deep learning, which pursue intuitive interpretations for practical applications, theoretical studies of deep learning never cease to seek fundamental understanding of its

---

[2]A review of existing eXplainable Artificial Intelligence (XAI) in deep learning, discussed using the taxonomy of ante-hoc and post-hoc methods along with their respective challenges, relevant to the topic of this research, is provided in Appendix A.2.

principles. These studies look for more universal explanations, such as the approximation capacities, learning behaviors, and knowledge generalizability of deep learning models, for which, various theories have been proposed.

One core theoretical framework explaining deep learning builds upon the Manifold Hypothesis, which posits that high-dimensional real-world data often lie on or near a lower-dimensional manifold (Tenenbaum et al., 2000; Roweis & Saul, 2000; Belkin & Niyogi, 2003). Neuroscience first identified similar principles in biological recognition systems (DiCarlo & Cox, 2007), later extended by Bengio et al. as a foundation for representation learning (Bengio et al., 2013). Brahma et al. formally advanced this hypothesis to explain deep learning effectiveness (Brahma et al., 2015), and subsequent evidence has further complemented the theory (Huang, 2018; Cohen et al., 2020):

**Theorem 1** (Manifold Learning in Deep Neural Networks). *Let $\mathcal{M} \subset \mathbb{R}^D$ be a $d$-dimensional data manifold ($d \ll D$), and $\mathcal{X} \subset \mathbb{R}^D$ be a finite set of (potentially noisy) samples drawn from a neighborhood of $\mathcal{M}$. Consider a deep neural network $f_\theta : \mathbb{R}^D \to \mathcal{Y}$ with parameters $\theta$, trained on $\mathcal{X}$ to approximate $g : \mathcal{M} \to \mathcal{Y}$. Then, $f_\theta$ learns a compositional map:*

$$f_\theta = \psi \circ \phi, \quad f_\theta \approx g$$

*where $\phi : \mathbb{R}^D \to \mathbb{R}^h$ ($h \geq d$) is a continuous map (often the penultimate layer) to a lower-dimensional Euclidean representation space $\mathbb{R}^h$ where $\mathcal{M}$ is approximately unfolded, and $\psi : \mathbb{R}^h \to \mathcal{Y}$ is a simple function (e.g., linear).*

This theorem implies that the internal representations learned by neural networks often exhibit lower-dimensional topological structures. Moreover, since these networks are primarily trained for regression or classification tasks, these structures are typically well-behaved and locally manifold-like (homeomorphic to a Euclidean space). We can therefore define such structures as a single or a collection of "learned manifold(s)":

**Definition 1** (Learned Manifold (Set) [3]). *For a deep neural network $f_\theta = \psi \circ \phi$ trained on samples near a data manifold $\mathcal{M} \subset \mathbb{R}^D$, a learned manifold $\mathcal{M}_\theta \subset \mathbb{R}^D$ is a single or collection of connected topological manifold(s) implicitly learned by the network through $\phi$ and is defined by the task:*

- *For a **regression** task: $\mathcal{M}_\theta$ is learned such that $\phi(\mathcal{M}_\theta) \approx \phi(\mathcal{M})$ in $\mathbb{R}^h$. It can be interpreted as the regression surface approximated by the set of connected sub-manifold(s) in the region where the mapping $\phi$ is locally injective.*
- *For a **classification** task with classes $\mathcal{K}$: a learned manifold $\mathcal{M}_{\theta:a,b}$ is defined for each pair of classes $(a, b)$ ($a \in \mathcal{K}, b \in \mathcal{K}$) as a or a set of connected sub-manifold(s) of their binary decision boundary. It is learned such that its embedding $\phi(\mathcal{M}_{\theta:a,b})$ lies on and approximates a hyperplane in $\mathbb{R}^h$ that linearly separates the representations of the two classes, $\phi(\mathcal{M}_a)$ and $\phi(\mathcal{M}_b)$.*

There are also many other theoretical works exploring how neural architecture—particularly the number of neurons—affects models' expressive power and representation capacity (Pascanu et al., 2013; Montúfar et al., 2014; Raghu et al., 2017; Lu et al., 2017; Hanin & Sellke, 2017a; Yarotsky, 2017; Park et al., 2020; Hanin & Rolnick, 2019), convergence behaviors and learning dynamics (Du et al., 2019; Jacot et al., 2018; Lee et al., 2019; Allen-Zhu et al., 2019), as well as optimization landscapes and generalizability (Belkin et al., 2019; Liang et al., 2018; Arora et al., 2019). A consensus from these studies indicates that increasing the number of neurons typically enhances a network's ability to approximate more complex functions. Geometrically, this process can be interpreted as locally increasing the topological complexity of the learned manifold by "folding" it on the input / feature spaces.

**Definition 2** (Folding on Manifold). *Given a neural network with parameters $\boldsymbol{\theta} \in \mathbb{R}^p$, let $\mathcal{M}_{\boldsymbol{\theta}} \subset \mathbb{R}^D$ be the maximum-complexity manifold (set) learnable by the network. Adding a neuron with activation $\sigma : \mathbb{R} \to \mathbb{R}$ transforms the learned manifold (set) into $\mathcal{M}_{\boldsymbol{\theta}'} \subset \mathbb{R}^D$ (with the new set of network parameters $\boldsymbol{\theta}' \in \mathbb{R}^q$, $q > p$), locally increasing topological complexity, which, geometrically performs a "folding" operation on the manifold / sub-manifold in form of either:*

- ***Non-Smooth Folding***: *If $\sigma$ is piecewise linear (e.g., ReLU), the manifold's complexity increases by introducing additional "crease" (non-differentiable edge):*

$$\mathcal{C}(\mathcal{M}_{\boldsymbol{\theta}'}) < \mathcal{C}(\mathcal{M}_{\boldsymbol{\theta}})$$

*where $\mathcal{C}$ denote a function of smoothness measuring the number of continuous derivatives (differentiability class) on the manifold.*

---

[3]Proof of the existence of learned manifold (set) are provided in the Appendix B.1

- **Smooth Folding**: *If $\sigma$ is smooth (e.g., sigmoid, tanh), the manifold's complexity increases by introducing additional variation to its total curvature:*

$$\int_{\mathcal{M}_{\theta'}} \|\kappa\|^2 dV \geq \int_{\mathcal{M}_{\theta}} \|\kappa\|^2 dV$$

where $\|\kappa\|^2$ *denotes the squared norm of the curvature tensor.*

Given that the mechanism of the GrowNNs algorithms is to expand network architecture through the strategic addition of neurons, we can interpret the process as follows:

**Theorem 2** (Growing Neural Networks as Folding Manifold [4]). *Network growth through the addition of a neuron induces a local folding of the learned manifold (set).*

Beyond interpreting neural network growth, a key innovation of the above framework is that: we can also precisely locate where "folding" occurs in input / intermediate feature spaces. This enables geometric analysis of GrowNNs algorithms through topological changes.

Specifically, each growth step can be formularized as follow:

**Definition 3** (Growing a Neuron). *Consider a Multi-Layer Perceptron (MLP), which is the most typical neural network, as a function $f_\theta : \mathbb{R}^D \to \mathcal{Y}$, taking an input $\mathbf{x} \in \mathcal{X}$ ($\mathcal{X} \subset \mathbb{R}^D$):*

$$f_\theta(\mathbf{x}) = \sigma_{L-1}\Big(\mathbf{W}_{L-1}\sigma_{L-2}\big(\cdots\sigma_1(\mathbf{W}_1\mathbf{x} + \mathbf{b}_1)\cdots\big) + \mathbf{b}_{L-1}\Big)$$

*where $L$ is the total number of layers, $\mathbf{W}_i$ and $\mathbf{b}_i$ denote the weight matrix and bias vector of the $i$-th layer, and $\sigma_i$ is the layer's activation function introducing non-linearity.*

*Growing a new neuron in the $i$-th layer of the MLP (where $1 \leq i \leq L-2$) appends new parameters to layers $i$ and $i+1$:*

$$\mathbf{W}_i^{new} = \begin{bmatrix} \mathbf{W}_i \\ \mathbf{w}_i^{new} \end{bmatrix}, \qquad \mathbf{b}_i^{new} = \begin{bmatrix} \mathbf{b}_i \\ b_i^{new} \end{bmatrix}, \qquad \mathbf{W}_{i+1}^{new} = \begin{bmatrix} \mathbf{W}_{i+1} & \mathbf{w}_{i+1}^{new} \end{bmatrix}$$

*where $\mathbf{w}_i^{new}$ (input-weight vector), $b_i^{new}$ (scalar bias), and $\mathbf{w}_{i+1}^{new}$ (output-weight column vector) are the new parameters added after growing each neuron.*

The spatial location of where "folding" occurs can be find by:

**Theorem 3** (Location of Folding[5]). *Given a neuron grown in the $i$-th layer of a neural network $f_\theta : \mathbb{R}^D \to \mathcal{Y}$, let $\mathbf{w}_i^{new}$, $b_i^{new}$, and $\mathbf{w}_{i+1}^{new}$ be the new parameters introduced on the $i$-th and the $(i+1)$-th layer by this growth, the folding of a learned manifold occurs on the $(D-1)$-dimensional pre-activation hyperplane under the linear constrain:*

$$\mathcal{F} = \{\mathbf{x} \in \mathbb{R}^D \mid \langle \mathbf{w}_i^{new}, \phi_{i-1}(\mathbf{x})\rangle + b_i^{new} = 0\}$$

*where $\phi_{i-1} : \mathbb{R}^D \to \mathbb{R}^{d_{i-1}}$ is the feature map up to layer $i-1$ ($\phi_0 = identity$). The output-weight vector $\mathbf{w}_{i+1}^{new}$ does not affect $\mathcal{F}$'s geometry.*

## 2.2 GROW BY FOLDING AT TYPICAL MISTAKES

Building on the geometric framework established above, we now address the second research question: How can we ensure the effectiveness of each growth step in Growing Neural Networks (GrowNNs)? Given that network growth is geometrically interpretable as folding the learned manifold (or manifold set) at specific locations, this question can be further specified as: Can we maximize growth effectiveness by identifying the optimal folding location during each growth step?

Before answering this question, we also need to define what is an "effective growth". The fundamental objective of growing a neural network is to enhance its capacity to approximate complex functions. Considerable theoretical studies also correlate increased neurons with reduction of systematic learning errors (Yarotsky, 2017; Hanin & Sellke, 2017b; Jacot et al., 2018). Thus, an effective growth can be considered as one that can mitigate systematic approximation errors arising from insufficient model capacity.

---

[4]Proof of this theorem is provided in the Appendix B.2.

[5]Proof of this theorem is provided in the Appendix B.3.

**Definition 4** (Typical Mistakes). *Given a neural network $f_\theta$ trained to approximate $g : \mathcal{M} \to \mathcal{Y}$, let $\mathcal{M}_\theta$ be a local manifold learned by the network. A "typical mistake" is an instance $\mathbf{x}_{mistake} \in \mathbb{R}^D$ such that:*

$$\mathbf{x}_{mistake} \in \mathcal{M}_\theta \wedge \mathbf{x} \notin \mathcal{M} \wedge \|f_\theta(\mathbf{x}_{mistake}) - g(\mathbf{x}_{mistake})\| > \tau$$

*where $\tau > 0$ is an error tolerance threshold.*

Geometrically, these systematic errors can be identified as clusters of prediction mistakes in the input space, based on the intuition:

**Hypothesis 1** (Clustering of Typical Mistakes). *Mis-predictions arising from the same model deficiency tend to distribute spatially close to each others in the input space.*

This motivates the core growth strategy of our GrowNNs method:

**Hypothesis 2** (Effective Growth by Folding at Typical Mistakes). *Effective neural network growth can be achieved by strategically folding the learned manifold at locations corresponding to the most frequent or severe approximation mistakes.*

### 2.3 FINE-TUNING FOR TARGETED MISTAKE CORRECTION

While identifying typical mistakes justifies the folding locations where network growth may be intuitively effective, the folding operation itself (i.e., adding a neuron) cannot directly correct mistakes. Instead, it is the subsequent fine-tuning after growth that exclusively mitigates these approximation errors. Therefore, measures also need to be designed to ensure post-growth fine-tuning effectively addresses these mistakes.

Recent studies (Evci et al., 2022; Yuan et al., 2023; Verbockhaven et al., 2024; Pham et al., 2024) also acknowledge the critical role of fine-tuning in GrowNNs, in contrast to classic GrowNN algorithms that focus solely on when and where to add new neurons. Specifically, these methods highlight the importance of initializing the new parameters that are introduced after adding neurons. For examples, the GradMax algorithm (Evci et al., 2022) proposes:

1. initializing incoming parameters for the new neuron $\mathbf{w}_i^{\mathrm{new}}$ and $b_i^{\mathrm{new}}$ in the prior layer to zero, so as to preserve the network output unchanged immediately after growth;
2. maximizing the gradient norm of the outgoing parameters $\mathbf{w}_{i+1}^{\mathrm{new}}$ in the later layer to promote rapid adjustment during fine-tuning.

However this approach has fundamental limitations:

- Regarding (1): as per our earlier findings, the new incoming parameters in the prior layer play a key role in defining the location where folding occurs and therefore cannot be set to zero. Conversely, setting the outgoing parameters $\mathbf{w}_{i+1}^{\mathrm{new}}$ in the later layer to zero would cause zero outputs from the new neurons, preventing gradients from being backpropagated through it;
- Regarding (2): GradMax only optimizes the norm of the gradient without considering its direction during fine-tuning. This results in an aimless optimization process.

To address these limitations, we first analyzed the roles of each new parameter before proposing our solutions to enforce fine-tuning towards effective correcting the identified mistakes.

**Theorem 4** (Outgoing Parameters Govern Output Displacement[6]). *For a neuron added to the $i$-th layer of a neural network $f_\theta$, with new parameters $\mathbf{w}_i^{new}$, $b_i^{new}$, and $\mathbf{w}_{i+1}^{new}$, let $\phi_{i-1}(\mathbf{x})$ denote the input and $\phi_{i-1}(\mathbf{x})$ be the feature passed to the $i$-th layer. The output perturbation is:*

$$\Delta f_\theta = \mathbf{w}_{i+1}^{new} \cdot \sigma(\mathbf{w}_i^{new} \cdot \phi_{i-1}(\mathbf{x}) + b_i^{new})$$

*where $\mathbf{w}_{i+1}^{new}$ determines the direction of output displacement, while $\mathbf{w}_i^{new}$ and $b_i^{new}$ control its magnitude through scaling after activation.*

Therefore, instead of setting $\mathbf{w}_i^{\mathrm{new}}$ to zero, we propose initializing $\mathbf{w}_{i+1}^{\mathrm{new}}$ as a non-zero uniform vector $\alpha \cdot \mathbf{1}$ ($\alpha \in \mathbb{R}$). This preserves non-zero gradients for backpropagation and avoids directional bias in output space (for classification tasks, it contributes equally to all classes, while for regression tasks, it also ensure controllable minimal shifts immediately after adding a new neuron).

---

[6]Proof of this theorem is provided in the Appendix B.4.

Accordingly, the idea of GradMax can be applied to initialize the remaining parameters $\mathbf{w}_i^{\text{new}}$. Notably, besides simply maximizing its gradient norm, we can also constrain its gradient direction in input space:

**Theorem 5** (Incoming Weights Govern Input-Space Gradient Direction[7]). *For a neuron added to the $i$-th layer of a neural network $f_\theta$, with new parameters $\mathbf{w}_i^{new}$, $b_i^{new}$, and $\mathbf{w}_{i+1}^{new}$, let $L(f_\theta(\mathbf{x}), \mathbf{y})$ be the loss function value, the perturbation of gradient in the input space is:*

$$\Delta \frac{\partial L}{\partial \mathbf{x}} = \left( \frac{\partial L(f_\theta(\mathbf{x}), \mathbf{y})}{\partial f_\theta(\mathbf{x})} \cdot \mathbf{w}_{i+1}^{new} \right) \sigma'(s) \cdot \mathbf{J}_{\phi_{i-1}}(\mathbf{x})^\top \mathbf{x}_i^{new}$$

*where $s = \mathbf{w}_i^{new} \cdot \phi_{i-1}(\mathbf{x}) + b_i^{new}$ and $\mathbf{J}_{\phi_{i-1}}(\mathbf{x})^\top$ is the Jacobian of $\phi_{i-1}$. For those activated region $s > 0$ in the input space, as long as the activation function is positive $\sigma$ (for example, for ReLU function, $\sigma'(s) = 1$ if $s > 0$), $\mathbf{w}_i^{new}$ dominates the change of direction for the networks' input-space gradient $\frac{\partial L(f_\theta(\mathbf{x}), \mathbf{y})}{\partial \mathbf{x}}$, $\mathbf{w}_{i+1}^{new}$ scales its magnitude, while $b_i^{new}$ does not affect its gradient.*

This enables our key innovation: since $\mathbf{w}_i^{\text{new}}$ controls the input-space gradient direction, we can initialize it to align the entire network's input-space gradient $\frac{\partial L(f_\theta(\mathbf{x})^{new}, \mathbf{y})}{\partial \mathbf{x}}$ with directions pointing toward the identified typical mistakes. Thus, the subsequent fine-tuning can explicitly optimize on the direction to correct these target mistakes.

Nevertheless, initialization alone cannot enforce the model's optimization behaviors during fine-tuning, thus it still cannot guarantee effective correction of the identified typical mistakes, resulting in the instability in the existing GrowNNs methods. To address this, a regularization can be easily proposed based on our geometric framework, to constrain the fine-tuning so that the folding hyperplane on the learned manifold can always be kept in the right direction:

**Definition 5** (Folding Hyperplane Regularization). *Let $\mathbf{w}_i^{new}$, $b_i^{new}$, and $\mathbf{w}_{i+1}^{new}$ be the parameters introduced by the newly grown neuron, $\mathbf{x}_{mistake}$ be the identified typical mistake that is used to fold and growth the new neuron, and $\mathbf{x}_{fold}$ be the original location where folding occurred on the original learned manifold in the input / feature space. During the fine-tuning process, an regularization $L_{fold\text{-}reg}$ can be applied by adding together distances from two points to the folding hyperplane:*

$$L_{fold\text{-}reg} = \frac{|\mathbf{w}_i^{new} \cdot \phi_{i-1}(\mathbf{x}_{mistake}) + b_i^{new}| + |\mathbf{w}_i^{new} \cdot \phi_{i-1}(\mathbf{x}_{fold}) + b_i^{new}|}{\|\mathbf{w}_i^{new}\|}$$

## 3  IMPLEMENTATION

Following the preceding analyses, we present our GrowNNs framework by answering the "when, where, how, why" questions below.

### 3.1  WHEN TO GROW:

Network growth is triggered only when model convergence is statistically verified. Following established practices (Wu et al., 2019; Evci et al., 2022), we define convergence as the state where further parameter updates yield no meaningful improvement in training loss. This ensures deficiencies in approximation capacity are identifiable once the model exhausts its existing expressive power, separating systematic errors from transient mis-predictions. Specifically, convergence is verified through two criteria: (i) attainment of a local minimum in training loss; and (ii) stationarity confirmed by an Augmented Dickey-Fuller (ADF) test (p-value $< 0.05$) applied to the losses over the last 20 epochs.

### 3.2  WHERE TO GROW:

The discussion on "where to grow" mainly concerns about the optimal layer for expansion in a deep network. Whereas, recent theoretical insights reveal that it may not need to be a concern: we can simply grow to expand the width in a shallow ReLU network. Considerable research (Hanin & Rolnick, 2019; Bietti & Bach, 2021; Mao & Zhou, 2023; Villani & Schoots, 2023) have indicated that, for ReLU-activated neural networks, increasing depth marginally impacts model expressiveness,

---

[7]Proof of this theorem is provided in the Appendix B.5.

while widening the network does. What even more astonishing finding is that: any deep ReLU network can be represented by a functionally equivalent shallow network. In this context, we propose commencing with a shallow ReLU-activated single-hidden-layer MLP and expanding it by incrementally adding neurons to this hidden layer [8]. For its expressiveness guarantees and simplicity, it is a universally adaptable baseline to grow from.

### 3.3 HOW TO GROW:

Our neural network growing strategy has been outlined in previous sections. Here, we consolidate the implementation through the following pseudocode of our main algorithm [9].

---

**Algorithm 1:** Grow by Folding at Mistakes

---

**Require:** compact single-hidden-layer ReLU-activated MLP $f_\theta : \mathbb{R}^D \to \mathbb{R}^Y$ with initial hidden size $h_0 \geq 1$; hidden weights $\mathbf{W}_A \in \mathbb{R}^{h_0 \times D}$, biases $\mathbf{b}_A \in \mathbb{R}^{h_0}$; output weights $\mathbf{W}_B \in \mathbb{R}^{h_0 \times Y}$, biases $\mathbf{b}_B \in \mathbb{R}^Y$; training set $\mathcal{X}$; loss $L$; regularization coefficient $\alpha > 0$ ;

$\mathbf{W}_A, \mathbf{b}_A, \mathbf{W}_B, \mathbf{b}_B \leftarrow$ train initial $f_\theta$ on $\mathcal{X}$ using $L$ until convergence;

**repeat**

    $\mathbf{x}_{\text{mistake}} \leftarrow$ cluster $f_\theta$'s mis-predictions in $\mathbb{R}^D$, find the centroid of the largest cluster,
           and identify the closest instance in $\mathcal{X}$ to centroid;

    **if** $\mathbf{x}_{\text{mistake}}$ not found **then**

        **return** $f_\theta$ with current parameters $\mathbf{W}_A, \mathbf{b}_A, \mathbf{W}_B$ and $\mathbf{b}_B$;

    **end if**

    $\mathbf{x}_{\text{fold}} \leftarrow$ find the closest point to $\mathbf{x}_{\text{mistake}}$ on the hyperplane of the corresponding local region
           of the learned manifold in $\mathbb{R}^D$ (regression surface or decision boundary);

    Initialize new neuron's parameters:

    $(\mathbf{w}_A^{\text{new}})^\top = \max_{\mathbf{w}_A^{\text{new}}}(\text{cosine-similarity}(\frac{\partial L}{\partial x}, (\mathbf{x}_{\text{fold}} - \mathbf{x}_{\text{mistake}})) + \mathbb{E}\|\frac{\partial L}{\partial \mathbf{w}_A^{\text{new}}}\|)$   `/* ensure the`
    `input-space gradient direction while maximize weight's gradient norm`         `*/`
    $\mathbf{b}_A^{\text{new}} = -(\mathbf{w}_A^{\text{new}} \cdot \mathbf{x}_{\text{fold}})$            `// ensure folding at` $\mathbf{x}_{\text{fold}}$
    $\mathbf{w}_B^{\text{new}} = \text{mean}(\mathbf{W}_B) \cdot \mathbf{1}$      `// initial outgoing parameters as a uniform vector`

    Grow by adding new neuron:

    $\mathbf{W}_A \leftarrow \begin{bmatrix} \mathbf{W}_A \\ (\mathbf{w}_A^{\text{new}})^\top \end{bmatrix}$

    $\mathbf{b}_A \leftarrow \begin{bmatrix} \mathbf{b}_A \\ b_A^{\text{new}} \end{bmatrix}$

    $\mathbf{W}_B \leftarrow \begin{bmatrix} \mathbf{W}_B & \mathbf{w}_B^{\text{new}} \end{bmatrix}$

    $\mathbf{W}_A, \mathbf{b}_A, \mathbf{W}_B, \mathbf{b}_B \leftarrow$ train current $f_\theta$ on $\mathcal{X}$ using $L + \alpha L_{\text{fold-reg}}$ until convergence;

**until** validation accuracy plateau

**return** $f_\theta$ with current parameter $\mathbf{W}_A, \mathbf{b}_A, \mathbf{W}_B$ and $\mathbf{b}_B$

---

### 3.4 WHY GROWTH IS EFFECTIVE:

Our GrowNNs framework allows justification of its rational with inherent (ante-hoc) interpretability: as each growth of a neuron targets a specific "typical mistake", which is an mis-predicted data instance identifiable within the dataset, conceivably in representative of a systematic approximation error to be addressed in the subsequent fine-tuning. By visualizing this instance of marginal case example, each neuron growth are tailored, can be justified and can be correspond to rectifying a specific verified deficiency of in-capable model, thus enable us to achieve an instance-based interpretation.

---

[8]While network depth remains pertinent to parameter efficiency and optimization dynamics (Mhaskar et al., 2017; Belkin et al., 2019), hence subsequent methods to rewrite grown shallow network into its deep equivariant and pruning the fully-connection MLP into sparsely-connected one will be discussed in later works given page limitation

[9]More implementation details are provided in Appendix C

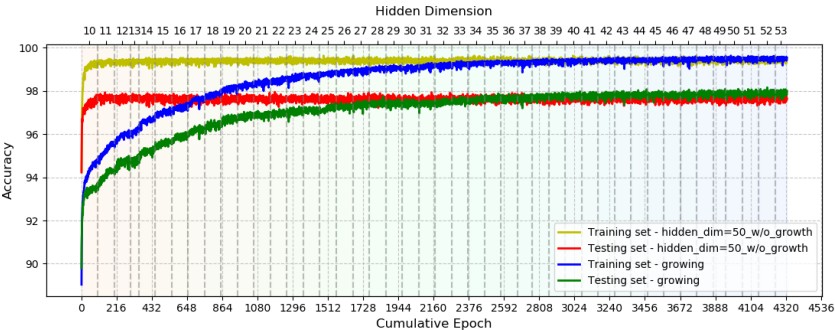

Figure 2: Changes of accuracies on training and testing set of the MNIST dataset on a network growth to 50 hidden neurons and compare to another network initialized to have 50 hidden neurons.

## 4 EXPERIMENT

To validate our framework, we first visualize the "folding" process by growing network model on some low-dimensional handcrafted manifolds. Specifically, we construct two datasets for simulation experiments: 1. classification of two sine functions (separable in the input dimension by non-linear function); 2. classification of two overlapped quadratic functions (inseparable in the current space). Both experiments demonstrated enhanced model approximation capabilities after a single growth of adding a neuron. For the sine classification task, we visually observed the folding process near identified "typical mistake" and corresponding adjustments of the decision boundary towards rectifying the mis-prediction during the subsequent fine-tuning process.

Besides the above directly visualizable empirical evidence in the low dimensional space, we also apply the proposed GrowNNs method on real-world dataset like the MNIST and CIFAR-10. Both dataset show continue increasing in accuracy after the growing of every single neuron until final convergence, indicating nearly the growth of every single neuron are effective in increasing the model's expressive power until they reach their learning ceiling on the dataset. On MNIST dataset, we can even find that the both models of the same architecture, the one grow from compact using our GrowNNs methods can achieve higher performance compare with another one that directly initalized with the large architecture, indicating higher parameter utilization during the growth process.

Table 1: Growing networks (to convergence by tests) vs. initially over-parameterized equivalent architectures on MNIST and CIFAR-10

| Dataset | Method | Testing Set Accuracy (mean $\pm$ sample standard deviation) |
|---|---|---|
| MNIST | growth to 50 hidden neuron | **97.7943 $\pm$ 0.1383** |
| | growth to 100 hidden neuron | **97.7948 $\pm$ 0.1380** |
| | initialized with 50 hidden neuron | 97.6459 $\pm$ 0.1156* |
| | initialized with 100 hidden neuron | 97.7806 $\pm$ 0.1205* |
| CIFAR-10 | growth to 70 hidden neuron | 50.4732 $\pm$ 0.6048 |
| | growth to 100 hidden neuron | 50.5286 $\pm$ 0.6028* |
| | initialized with 70 hidden neuron | 50.4090 $\pm$ 0.5970* |
| | initialized with 100 hidden neuron | **50.9484 $\pm$ 0.6067*** |

* indicates significant difference with p-value $< 0.0001$ in t-test compare with the first method

A key advantage of our proposed framework is the interpretability it provides by visualizing the the corresponding "typical mistake" identified in each growth step. Figure 3 demonstrates the identified mis-predicted data instance when training on the MNIST dataset. Most of these identified typical mistakes are, indeed, marginal cases that are easily confused between the wrongly predicted class and the ground truth.

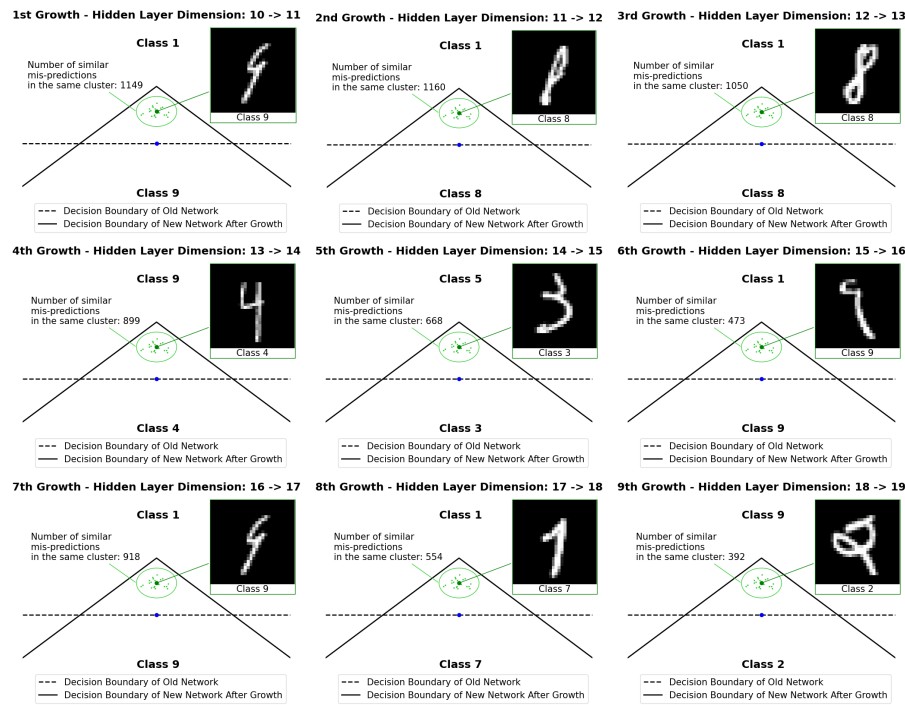

Figure 3: Visualization of the identified typical mistakes in the first 9th growth on MNIST dataset.

## 5 DISCUSSION

This study introduces a foundational framework for eXplainable Network Growth (XNG), which interprets the growth of every single neuron in a neural network as a process of "folding" a learned manifold to address a model's systematic prediction mistake. Relevant results validate that the proposed framework can incrementally grow a network to achieve performance competitive with, and sometimes superior to, a statically-initialized model of the same larger architecture. This enables identifying an appropriate model scale during growth, moving beyond the conventional practice of initializing an over-parameterized network based on empirical guesswork. Furthermore, the proposed framework provides ante-hoc explainability for the growth process. Each added neuron is justified by a specific deficiency in the model's current capability, which can be directly observed by visualizing the corresponding mis-predicted data instance that the neuron grown targeted at.

The GrowNNs framework presented in this study is currently restricted to width expansion on the single hidden layer of a shallow, fully-connected network (MLP). However, network depth and layer sparsity play critical roles in performance and parameter efficiency. Promisingly, findings from recent studies suggest that a shallow-to-deep conversion and eXplainable Network Pruning (XNP) could be adopted after the growth process to further enhance learning. These extensions, while important, are beyond the scope and page constraints of this paper and are left for future work.

While the growth process can still introduces overfitting, the explainable nature of our framework offers a unique solution: it allows for human intervention to guide the network growth via early stopping or by rolling back inappropriate growth steps. This paves the way for eXplainable Network Editing (XNE) with interactive human control, an aspect we will also elaborate on in future studies.

Finally, compared to adding a single neuron at a time, growing multiple neurons simultaneously sacrifices some parameter efficiency but allows learning more complex features. This is illustrated by our CIFAR-10 results: although growth to 70 neurons outperformed a statically-initialized 70-neuron model, an over-parameterized model with 100 neurons achieved higher performance than one grow to 100, suggesting that extracting complex features may require multiple neurons' addition. Thus, investigating the optimal growth granularity—balancing efficiency and feature extraction—is a valuable future direction.

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
