# OpenReview forum: "Growing Networks by Folding Manifolds at Mistakes"
_ICLR.cc/2026/Conference — Submitted to ICLR 2026_

### Official Review · Reviewer_Njy8 · 2025-10-28

**Soundness:** 2
**Presentation:** 3
**Contribution:** 3
**Rating:** 4
**Confidence:** 3

**Summary:**

Current Growing Neural Networks (GrowNNs) lack interpretability. The authors argue that the performance improvement observed in each growth step of GrowNNs arises from a process of geometrically folding a learned representation manifold. They further propose that the most effective growth occurs when the folding takes place in manifold regions associated with the model’s typical mistakes. The authors systematically apply this manifold-based perspective to the growth mechanism and strategy of GrowNNs, providing formal definitions and theoretical proofs. Finally, experiments conducted on the MNIST and CIFAR-10 datasets demonstrate the effectiveness of their proposed growth strategy.

**Strengths:**

1. The paper introduces the concept of a manifold and provides an innovative geometric interpretation of the GrowNNs growing process, effectively addressing the lack of ante-hoc interpretability in this field.

2. The authors propose a complete ante-hoc execution strategy for GrowNNs, covering the questions of when, where, and how to grow. Each component is formally defined and theoretically proven, lending strong credibility to the framework.

3. The paper is well organized and logically progressive. It first presents the manifold perspective (Theorem 1), then introduces the definitions and theorems of Learned Manifold, Folding, and Growing a Neuron (Definition 1–3, Theorem 2–3), and finally proposes the strategy of folding at typical mistake regions.

**Weaknesses:**

1. The experimental section appears relatively limited compared to the theoretical part. For example:

    - The paper lacks comparisons with other growth strategies, making it difficult to determine whether performing growth at "typical mistake" regions is superior to alternative growth approaches.

    - Subsection 3.4 reads more like a theoretical explanation rather than empirical validation. Ablation studies would be valuable to clarify why the proposed method is effective. Specifically, the growth strategy involves carefully designed components for typical-mistake selection, loss and folding regularization, and initialization of new parameters, yet there are no ablation experiments to quantify the independent and combined contributions of these components to overall effectiveness.

2. The experiments and interpretability analyses focus only on the neuron growth process within an MLP, without extending the framework to other types of neurons such as convolutional neurons. Nevertheless, this paper remains a strong contribution to the GrowNNs field.

3. Most figures and tables are not explicitly referenced or discussed within the relevant text, which may make it difficult for readers to understand. In addition, several typos were found in Appendix B2.2.

**Questions:**

1. In Figure 3, the changes in the decision boundary before and after growth are not clearly visible. Is this figure merely an illustration, rather than a visualization of actual experimental results?

2. According to Table 1, when comparing networks with the same number of neurons, the performance improvement of the grown network over the initialized network appears limited. However, Figure 2 shows that before the network grows to 50 neurons, its performance already surpasses that of a statically initialized 50-neuron model. Does this imply that under this growth strategy, better performance can be achieved with fewer neurons? A more detailed presentation of such phenomena, as in Figure 2, might be what readers would most expect to see.

---

### Official Review · Reviewer_6oUo · 2025-10-31

**Soundness:** 3
**Presentation:** 3
**Contribution:** 2
**Rating:** 4
**Confidence:** 3

**Summary:**

This paper introduces a new method for growing single-layer ReLU networks: It interprets adding a neuron as a "folding" of the decision boundary, and then aims to introduce these folds close to points that are misclassified by the network (adding capacity exactly where needed). The resulting algorithm performs well compared to a static (non-growing) network.

**Strengths:**

* Novel approach to growing
* The resulting algorithm has nice interpretability results: Each growth operation can be directly linked to a cluster of misclassified examples and a single representative example of this cluster whose misclassification this new neuron aims to address.

**Weaknesses:**

* Experiments do not compare against other growing methods (GradMax, Firefly, random growth, etc.)
* Theory and algorithm only apply to single-layer ReLU networks. No trivial extension to other more complex (real-world) networks such as multi-layer MLPs, convolutions, etc. This is worrisome because in the deep network case it becomes non-trivial to find examples that lie closest to the decision boundary.
* Scaling issues: Each growth operation requires embedding the full training set, clustering a large dataset (of potentially high-dimensional points, depending on the input space--this also assumes that the input space is amenable to clustering) and then solving the complicated problem of finding the closest point to the decision boundary.

**Questions:**

Do the authors have a path in mind towards adapting this method to actual real-world applications: multi-layer networks/CNNs/RNNs/etc., activation functions other than ReLUs, high-dimensional inputs, etc. And do you have any intuition as to whether this would actually be practical?

---

### Official Review · Reviewer_5VtH · 2025-10-31

**Soundness:** 3
**Presentation:** 2
**Contribution:** 2
**Rating:** 2
**Confidence:** 4

**Summary:**

This work proposes a framework for training GrowNNs, neural networks that increase their representational capacity by adding neural progressively. This is in contrast to existing approaches that make networks as large as possible, then potentially prune them. The proposed approach works by identifying points that are mispredicted, then "folding" the learned manifold nearest to these points by adding a new neuron, then fine-tuning the new parameters.

**Strengths:**

1. The idea of progressively growing networks, though not novel, is important for building efficient models.
2. The algorithm is rigorously defined and supported by theory.
3. Though not emphasized in the text, the potential savings in terms of parameters in Figure 2 is significant.

**Weaknesses:**

1. Limited evaluation focused on toy models and datasets.
2. Missing comparisons to other "grow" frameworks.

**Questions:**

Q1: How do you compare in terms of final performance and size of network to other GrowNN frameworks (e.g. GradMax or [1])?

Q2: The approach detailed (especially in the 2-layer MLP case) seems related to prior work on spine theory for NNs [2]. Could you comment on the connection, and potentially cite if it is relevant?

Minor comments:
- Theorem 1: definition of Y is missing
- Line 197: typo "constrain" --> "constraint"
- Definition 4: should it be x_{mistake} \not \in M ?
- Figure 1 is never cited in the text

[1] : https://proceedings.neurips.cc/paper/2020/hash/fdbe012e2e11314b96402b32c0df26b7-Abstract.html
[2] : https://www.arxiv.org/abs/1910.02333v1

---

### Meta-Review · Area_Chair_4rRB · 2026-01-12

**Summary:**

This paper proposes a framework for training Growing Neural Networks by interpreting neuron addition as a geometric “folding” of a learned manifold near typical mistakes. The authors provide formal definitions and theoretical results, and demonstrate the approach on simple MLPs with experiments on MNIST and CIFAR-10, reporting improved performance compared to static networks.

The paper’s strengths lie in its clear geometric interpretation of growth, rigorous theoretical formulation, and the appealing idea of adding capacity where errors concentrate, which reviewers found interpretable and conceptually interesting (5VtH, 6oUo, Njy8).

However, the submission has significant limitations. Most reviewers point to the very limited empirical evaluation: experiments are restricted to shallow networks and small benchmarks, with no comparisons to established GrowNN methods such as GradMax or Firefly (5VtH, 6oUo, Njy8). The lack of ablation studies makes it unclear which components of the proposed growth strategy actually drive gains (Njy8). Moreover, the theory and algorithm apply only to single-layer ReLU networks, with unclear scalability and practicality for deeper or real-world architectures (6oUo). These gaps substantially weaken the paper’s impact.

**Reviewer Concerns:**

The most critical weakness is the absence of comparisons to existing growth frameworks. Reviewers 5VtH and 6oUo noted that without benchmarks against methods like GradMax or Firefly, it is impossible to determine if this manifold-based strategy offers a tangible advantage. Furthermore, the evaluation is largely restricted to toy models and single-layer ReLU networks. Reviewer 6oUo expressed significant concerns regarding the scalability of the approach, specifically the computational overhead of clustering high-dimensional data and finding decision boundary proximities in deeper architectures. Additionally, Reviewer Njy8 pointed out a lack of ablation studies and several presentation oversights, such as unreferenced figures.

Given the narrow architectural scope and the lack of comparative evidence against state-of-the-art growth strategies, the paper is not recommended for acceptance at this time.

**Reviewer Scores:**

It is hard to tell. The authors did address some of the concerns.

---

### Decision · Program_Chairs · 2026-01-26

Reject